# Environmental drivers of stream metabolism in a middle TN headwater stream

**Ming Chen** [ID] [◉], **John C. Ayers** [ID]*[◉]

Department of Earth and Environmental Sciences, Vanderbilt University, Nashville, Tennessee, United States of America

◉ These authors contributed equally to this work.
* john.c.ayers@vanderbilt.edu

**Data Availability Statement:** All code, photographs, and model input and output files used in this study are publicly available in the following Figshare repository: Chen, Ming; Ayers, John (2024). EFC_Stream_Metabolism. Figshare.

## Abstract

Monitoring the seasonal and diurnal variations in headwater stream metabolic regimes can provide critical information for understanding how ecosystems will respond to future environmental changes. In East Fork Creek, a headwater stream in middle Tennessee, week-long field campaigns were set up each month from May 2022 to May 2023 to collect stream metabolism estimators. In a more extensive field campaign from July 2–5 in 2022, diel signals were observed for temperature, pH, turbidity, and concentrations of Ca, Mg, K, Se, Fe, Ba, chloride, nitrate, DIC, DO, DOC, and total algae. Gross Primary Productivity (GPP) and Ecosystem Respiration (ER) were calculated based on a Bayesian model using the dissolved oxygen (DO) time series approach. DO showed diurnal swings between oversaturation in daytime and undersaturation at night, with DO amplitudes being greatest in summer. GPP measurements have a clear seasonal variation, peaking in July and staying low in winter, and strong diel signals that couple with the daily light regime variation. ER does not vary seasonally except for a slight increase in Fall which might be caused by terrestrial organic inputs. The dominant control on GPP is light intensity and on ER is temperature. East Fork Creek shows a heterotrophic metabolic regime for 54 of 57 campaign days and therefore consumes $O_2$ and emits $CO_2$ to the atmosphere throughout the year. If carbon inputs are not a limiting factor, the positive temperature dependence of ER may cause increased $CO_2$ emissions from headwater streams and more frequent hypoxia events in a warming climate.

## Introduction

Stream metabolism consists of gross primary productivity (GPP), ecosystem respiration (ER), and net ecosystem productivity (NEP), such that NEP = GPP + ER where ER is a negative flux driven by autotrophic and heterotrophic respiration [1]. Photosynthesis dominates during the day, consuming $CO_2$ and releasing $O_2$, and contributing to GPP. ER occurs throughout the day but dominates at night, resulting in net $CO_2$ emissions at night. GPP, ER, and NEP are expressed as oxygen fluxes (with units like g $O_2$ m$^{-2}$ d$^{-1}$), but they will be strong indicators to show if the stream is accumulating or emitting carbon [1] since $O_2$ and $CO_2$ concentrations are negatively correlated. The stream system is considered autotrophic if NEP is positive and heterotrophic if NEP is negative [2].

Collection. https://doi.org/10.6084/m9.figshare.c. 7525866.v3. Summary data files for stream metabolism, which form the basis of many figures in the manuscript, are stored in: Ayers JC and Chen M (2024). Stream metabolism study of East Fork Creek, Franklin, TN in years 2022-2023 ver 1. Environmental Data Initiative. https://doi.org/10. 6073/pasta/4a7e714cd0395e1dafecfe49c7fdf84b.

**Funding:** This research was supported by the Alberstadt-Reesman-Stearns Fund from the Earth and Environmental Sciences Department at Vanderbilt University, awarded to MC. The funders had no role in study design, data collection and analysis, decision to publish, or preparation of the manuscript.

**Competing interests:** The authors have declared that no competing interests exist.

In a clearwater shallow headstream, the benthic population can be the dominant primary producer in the ecosystem [3,4]. In streams, benthic productivity is found to be sensitive to multiple environmental attributes such as light intensity, nutrients, temperature, and tree coverage which can vary significantly throughout a year or even a day [5]. Hence, measurements of diurnal and seasonal changes in benthic productivity can lead to meaningful insights into the nested result of changes in those environmental attributes.

The interaction between GPP and ER and gas exchange with the atmosphere control the amount of available dissolved oxygen (DO) in the stream [5], so measurements of DO and NEP will be an important indicator of hypoxia events in the stream. When the stream constantly experiences negative NEP (GPP $<$ |ER|) and low gas exchange rate, the stream will have frequent hypoxia, putting the stream's health at risk [6].

Earlier studies on stream metabolism explained the spatial distribution of metabolic regime variations along a river. The proportion of $CO_2$ from terrestrial fluxes of organic carbon decrease and of aquatic metabolism increase downstream from the headwaters [7]. Even 1st order streams can be more heterotrophic than 2nd order streams in the same watershed due to increasing GPP downstream [8].

Stream metabolism is directly affected by light intensity, temperature, and nutrient availability. Light intensity is a dominant control on GPP [9], so any factors that affect light intensity such as seasonality, canopy cover, and turbidity will indirectly affect GPP. Temperature has a strong influence on ER [10] and likely affects GPP, although the positive correlation between light intensity and temperature makes it difficult to quantify the dependence of GPP on temperature [9]. While GPP should increase with increasing concentration of the limiting nutrient (usually phosphorous in freshwaters) as it does in lakes [11], GPP and ER often show little or no dependence on nutrient concentrations in streams [12], so the effect of nutrient concentrations was not a focus of this study.

Few studies have continuously monitored stream metabolism in headwater streams and explained the effects of light intensity and temperature on stream metabolism on an annual scale. A study of streams in Queensland AU found that stream metabolism was affected by light intensity, phosphorous concentration, temperature, and forest cover, but measurements were made for only one month [5]. Continuous measurements of stream metabolism in a headwater stream over a two-year period showed that GPP and ER were strongly correlated with light intensity and water temperature [13]. GPP was highest during summer months, while ER increased with water temperature throughout the year. Similarly, a long-term study of stream metabolism in a forested headwater stream found that GPP was primarily limited by light availability, with peak values occurring during the summer, and ER was positively correlated with water temperature, with the highest rates occurring during the warmest months [14]. Another study examined the seasonal patterns of stream metabolism in a small headwater stream and found that GPP was highest during the summer when light levels were maximized, while ER showed a positive relationship with water temperature throughout the year [15]. These studies demonstrate the importance of light intensity and water temperature in regulating the annual patterns of GPP and ER in headwater streams.

The objective of this research was to measure the diurnal and seasonal variations of stream metabolism of East Fork Creek in Middle Tennessee using DO time series and identify potential environmental controls of stream metabolism. Environmental factors that affect stream metabolism directly (light intensity and temperature) and indirectly (turbidity and seasonality) were measured. As a rural headwater stream, data from East Fork Creek (EFC) can provide valuable insights on how stream metabolism responds to changes in environmental factors with few human disturbances. Then, based on the data from EFC, we tested stream metabolism parameter relationships established by previous studies. Research questions pertaining to

headwater streams include: How do ER and GPP depend on temperature [10]? How does GPP vary with light intensity? How does stream metabolism vary seasonally? Is EFC consistently heterotrophic (a $CO_2$ source) or is it sometimes autotrophic (a $CO_2$ sink)? How do measured DO concentrations compare with predictions made by the established temperature- and pH-dependent scaling law for freshwater streams in the USA [16]? Is there any risk of hypoxia in EFC [6]? The answers to these questions will help us better understand how climate change will affect stream metabolism and the carbon and oxygen budgets for headwater streams.

## Materials and methods

### Study sites

The focus of this study was East Fork Creek, an open-canopy headwater stream in a rural setting in Franklin, TN (Fig 1). The stream is 22.4 km long with a watershed coverage of 7.54 km$^2$ [17]. In Franklin County, where EFC is located, the bedrock is predominantly the Ordovician Nashville Group, which is composed of the Catheys Formation, Bigby-Cannon Limestone, and Hermitage Formation [18]. Weathering of these phosphate-rich limestones can form phosphatic soil that can potentially serve as a carbon source for the stream [19].

The stream monitoring was conducted in two sites on land owned by the Center for Sustainable Stewardship, which approved our study. These sites are downstream from Stephens Lake, a 0.026 km$^2$ pond formed by damming the headwaters of EFC (Fig 1). Previous research on the effect of small dams on DO in Massachusetts, USA showed no consistent effect, i.e., downstream DO was similar to upstream DO [20]. Site 1 (35.95157˚N, 87.01615˚W) was used to study diurnal variations in EFC in July 2022 and for stream metabolism measurements from May to October 2022. Site 1 is 1290 meters downstream from Stephens Lake and has a wide river channel with no leaf coverage all year and easy access from the road. Site 2 (35.960180˚N, 87.015930˚W), used for most stream metabolism measurements, was located 2240 meters downstream from Stephens Lake. Site 2 is a more protected site near a bridge with a narrower channel and sparse leaf coverage. Artificial light is mostly absent from the stream valley (Bortle Class 4 "rural to suburban transition"), so nighttime measured light intensity was usually zero.

**Field campaign setup.** From May 2022 to December 2023, field campaigns were carried out almost every month. In regular field campaign days, we only set up water quality sensors to collect data for stream metabolism calculations based on DO time series. The measurements for input parameters were made using a miniDOT dissolved oxygen and temperature logger, a HOBO Pendant MX2202 Temperature/Light Data Logger, and an Onset HOBO U20L-01 Water Level Data Logger. Detailed instructions are provided in S8 File. We confirmed that the MiniDOTs and HOBO Pendant MX Temperature/Light Data Loggers serve well for the purpose of this study (S1.1 –S1.4 in S1 File).

Each month measurements were made every 5 minutes for 3–7 days. PAR was calculated using measured illuminance in lux times a constant conversion factor of 0.0185 [21]. Simultaneous measurements made using a LI-192 underwater PAR sensor acquired late in this study were on average 23% higher. Based on previous measurements in EFC, the salinity was around 0.12 ppt with minor variation. Using 0.25 ppt as input caused negligible differences in the result. Therefore, we take the value of 0.12 ppt for salinity as the constant input for all our models. Sometimes we would deploy a YSI EXO2 sonde to measure DO and compare with the measurements from the miniDOT (S1.1 in S1 File) and found good correlation (Pearson r = 0.997).

To study diurnal variations from July 2nd to July 5th, 2022, along with water quality sensors, we deployed an ISCO auto-sampler and YSI EXO2. The YSI EXO2 was set to measure

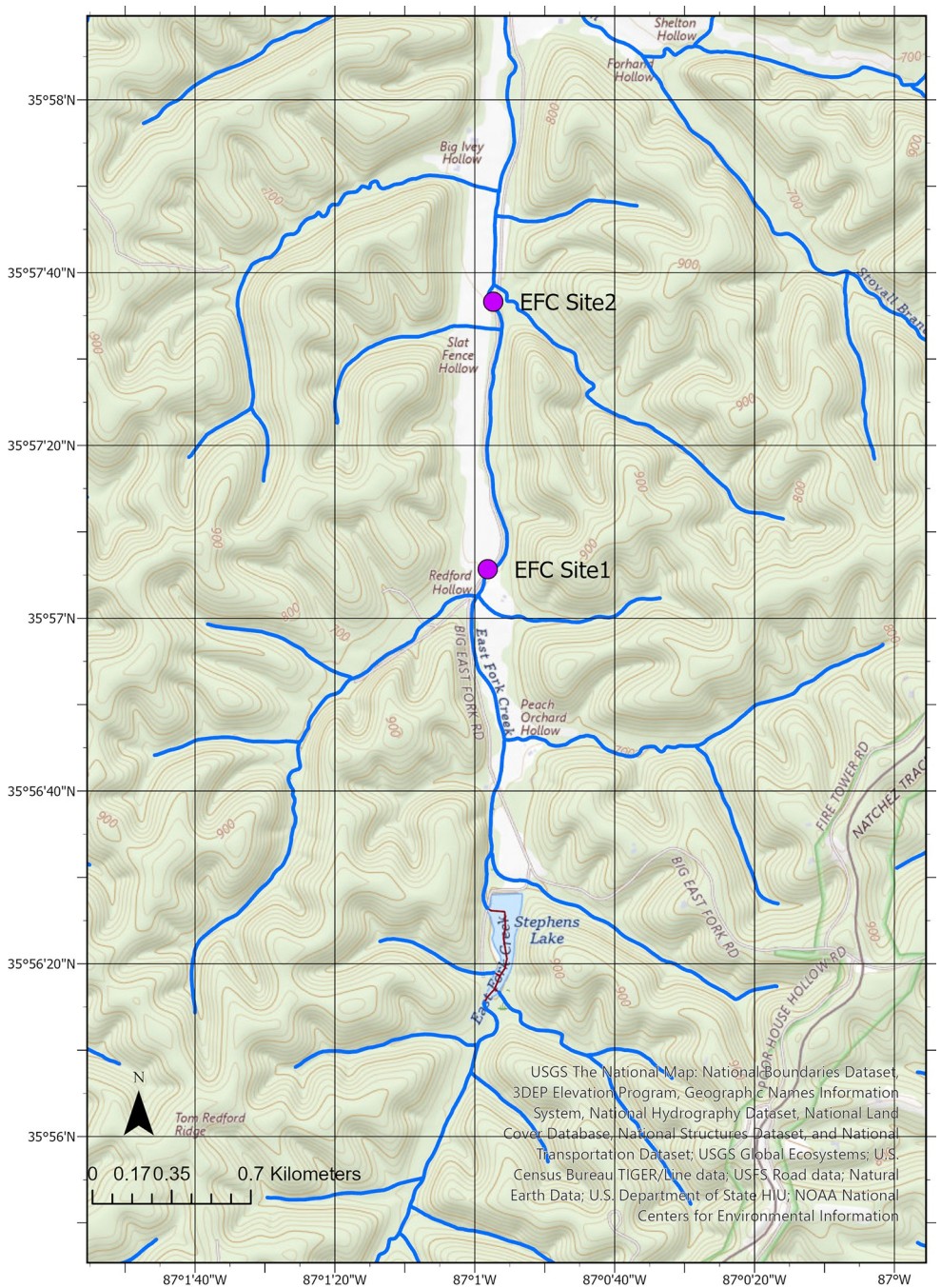

**Fig 1. USGS topographic map with hydrology (National Hydrologic Dataset flowlines) and study site locations.**
From The National Map (https://www.usgs.gov/programs/national-geospatial-program/national-map). Data available from U.S. Geological Survey, National Geospatial Program.

specific conductivity (uS/cm), depth (m), dissolved oxygen (mg/L), fluorescent dissolved organic matter (fDOM in RFU), pH, total algae (RFU), and turbidity (FNU) every 5 minutes. Before the campaign we calibrated the EXO2 sonde for conductivity, DO, pH, and turbidity. Later we temporally paired the EXO2 measurements and BASEmetab outputs with the analyzed species concentrations in collected water samples.

**Water sample collection and analysis.** For the July 2022 study of diurnal variations, we collected water samples using an ISCO auto-sampler. The auto-sampler was set to collect 150ml water samples from EFC every four hours, for 24 samples. Samples were stored inside the auto-sampler with ice during the field campaign and then refrigerated upon return to Vanderbilt University.

Sample preparation procedures follow Dietrich and Ayers [22,23]. Samples were processed immediately after the last sample was collected. Each water sample was filtered into two 30 mL centrifuge tubes using a 0.45 μm Whatman ashless paper filter and a syringe. One set of 30 mL filtered samples was acidified by adding 300 μL $HNO_3$ to make 1% $HNO_3$ for inductively coupled plasma analysis (ICP-MS and ICP-OES). The other set of water samples was used for ion chromatography (IC) and total organic carbon analysis (TOC).

All acidified samples were analyzed using ICP to provide concentrations in μg/L for different chemical elements. Ca, Fe, Mg, Mn, P, K, Si, Na, and S concentrations were measured using a Varian ICP Model 14720-ES ICP-OES following EPA Method 6010D, and Sb, As, Ba, Be, Cd, Cr, Cu, Pb, Se, Ti concentrations were analyzed using a Perkin Elmer model ELAN DRC II ICP-MS in both standard and dynamic reaction chamber modes using EPA Method 6020B. Five milliliters of unacidified samples were analyzed for fluoride, chloride, nitrite, bromide, nitrate, phosphate, and sulfate concentration on a Metrohm 881 Compact IC pro using American Society for Testing and Materials (ASTM) Method D-4327-03. Inorganic carbon and total carbon concentration used 15 mL of unacidified water samples, and the analyses were performed on a Shimadzu model TOC-V CPH/CPN using ASTM Method D-7573-09.

**Data analysis: Stream metabolism.** All data analysis and visualization was conducted using the R programming language v. 4.4 [24] and the RStudio integrated development environment [25]. The stream metabolic patterns were analyzed using our DO time series measurements and the Bayesian Single-station Estimation (BASEmetab) package in R. To use this single station model, the stream needs to satisfy the following characteristics: (a) unidirectional flow, (b) narrow and shallow river channel with a consistent flow indicating the stream is well mixed laterally and with depth, (c) homogeneous distribution of GPP, ER, and K along the length of the stream, (d) no evidence of groundwater input or wastewater disposal that can serve as external source of oxygen, and (e) a strong diel signal in DO and GPP measurements [26]. Based on the field observations and experiments, we found that EFC sites 1 and 2 are suitable sites for a 1-station model analysis of stream metabolism.

The BASEmetab model makes stream metabolism inferences using Bayesian model fitting based on periodic measurements of dissolved oxygen (DO) concentrations, water temperature, light intensity, and atmospheric pressure [27]. The time rate of change of DO concentration at time increment $i$ is affected by processes represented by three terms on the righthand side of Eq (1), which from left to right are GPP, ER, and gas exchange with the atmosphere:

$$\Delta[DO]_i/\Delta t = AI_i^p - R(\theta^{(T_i-T)}) + K_{O_2}(1.0241^{(T_i-T)})D_i \qquad (1)$$

The constant A represents the primary production per quantum of light, exponent $p$ is the coefficient for light saturation and represents the primary producer's ability to use incident light, R represents the rate of ER, $T_i$ is the water temperature, T is the mean water temperature for that day, and $\theta^{(T_i-T)}$ accounts for the temperature dependence of the respiration rate [27]. The constant θ is set to a default value of 1.072 and $p$ to a value of one. $K_{O_2}$ is the reaeration coefficient ($d^{-1}$), and $Di$ is the $O_2$ saturation deficit/surplus (mg $O_2$ $L^{-1}$), which is affected by temperature, atmospheric pressure, and salinity. The inputs for this model are DO concentration [DO] (mg L-1), irradiance ($I$) (PAR = photosynthetically active radiation expressed as photosynthetic photon flux density in umol $m^{-2}$ $s^{-1}$), water temperature (˚C), atmospheric

pressure (atm), and salinity (ppt = parts per thousand). Models were run with and without priors, with little difference in results.

The model output includes estimated 5-minute and daily average values for GPP, ER, NEP, and the reaeration coefficient for $O_2$ ($K_{O_2}$). The 5-minute measurements are over 5-minute increments between measurements, while the daily average values are obtained by summing the 5-minute values for that day. The posterior predictive p-value was used to perform quality checks on output data. Values around 0.5 indicate plausible results, while values larger than 0.9 or smaller than 0.1 are implausible and should be rejected [27]. Based on this criterion, all our model results are acceptable.

## Results

Based on the water sample analysis results from the July, 2022 field campaign at site 1, EFC is Ca-HCO$_3$ water type (S3.1 Fig in S3 File). Other species concentrations were much lower than the dominant species, $Ca^{2+}$ and $HCO_3^-$. None of the measured species' concentrations exceeded U.S. regulatory limits (S3.1 Table in S3 File). All reported values of GPP, ER, and $K_{O_2}$ are model outputs from BASEmetab.

### Diurnal variations: July 2022

Data and water samples collected at site 1 from July 2nd to July 5th, 2022, including the geochemical analysis results from the water samples and stream metabolism measurements, were used to examine the diurnal variations in EFC's metabolic regime. Time series plots for chemical concentrations were generated based on 24 discrete data points where water samples were collected every 4 hours (Fig 2). Continuous time series plots were generated from EXO$_2$ and HOBO light logger measurements made every 5 minutes (Fig 3). The diurnal variations in Ca, Mg, and pH showed a pattern closely related to the changes in GPP and ER. Other parameters that showed diel signals included temperature, turbidity, and concentrations of K, Se, Fe, Ba, chloride, nitrate, DO, DOC, and total algae (TAL). Time series plots for all measured chemical species and parameters are in (S4.1 & S4.2 Figs in S4 File).

Spearman correlation analysis was used to determine the correlation matrix. Statistically significant correlations (p < 0.05) with r values > 0.5 are reported here for July 2022 at site 1. For discrete measurements (S5.1 Fig in S5 File) GPP was correlated with fDOM (-0.65), DO (0.88), TAL (-0.60), pH (0.78) and concentrations of Ca (0.66), Ba (0.5) and nitrate (0.65). ER was correlated with temperature (0.77), $K_{O_2}$ (0.81), SpC (-0.78), TAL (-0.71), pH (0.61) and nitrate (0.52). For continuous measurements (S5.2 Fig in S5 File) GPP was correlated with light intensity (0.90), temperature (0.51), pH (0.56), DO (0.64), and DOM (-0.51), and ER was correlated with light intensity (0.57), $K_{O_2}$ (0.81), temperature (0.80), pH (0.63), SpC (-0.61), DO (0.74), and TAL (-0.76). Measured DO concentrations ranged from 5.7 to 11.1 mg/L (67% to 143% local saturation), indicating an aerobic environment, and the mean P concentration of 18 ug/L indicates the stream was mesotrophic.

### Seasonal variations

In October 2022 our equipment was stolen from site 1, so all subsequent measurements were made at site 2 located ~ 1 km downstream from site 1. Site 2 was also less likely to be affected by water storage in the lake. Data collected simultaneously at the two sites in October 2022 showed that GPP, ER, and $K_{O_2}$ values were highly correlated (S2 File). Among the 57 days of measurements from October 2022 to December 2023, 22 were collected at site 1.

Measurements throughout the year were grouped by seasons, spring (March to May), summer (June to August), fall (September to November), and winter (December to February), to

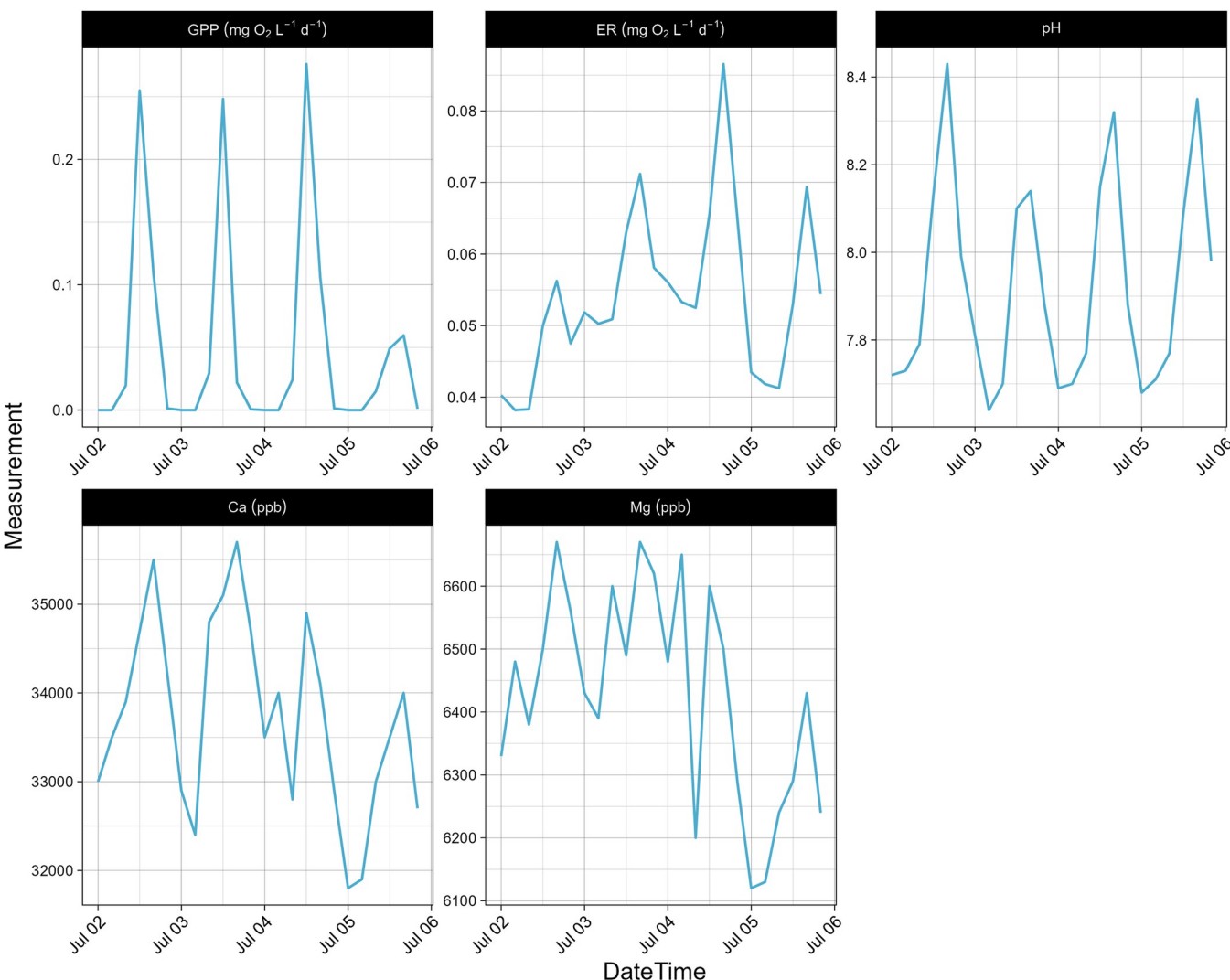

**Fig 2. Time series for discrete measurements from EXO$_2$ and water samples at East Fork Creek site 1.** The samples were collected every four hours from July 2$^{nd}$ to 6$^{th}$, 2022 for geochemical analysis. Ca and Mg concentrations are in ppb.

examine the seasonal variations. Only on four of the 57 days did GPP exceed or equal |ER| (3 in winter and 1 in summer), indicating that EFC was usually heterotrophic throughout the year (Fig 4). GPP showed strong seasonal variation, peaking in July (18 mg O$_2$ L$^{-1}$ d$^{-1}$) and staying low in winter with a median of 3. 8 mg O$_2$ L$^{-1}$ d$^{-1}$ (Figs 5 and 6). |ER| did not show strong seasonal variation but reached two small peaks in late spring and fall (Fig 6). The median values for ER were -17 mg O$_2$ L$^{-1}$ d$^{-1}$ in summer, -9.2 mg O$_2$ L$^{-1}$ d$^{-1}$ in winter, but -25 mg O$_2$ L$^{-1}$ d$^{-1}$ in late spring (measurements in May) and -22 mg O$_2$ L$^{-1}$ d$^{-1}$ in fall (Fig 5). Net ecosystem production (NEP) = GPP + ER was usually negative, since ER is the dominant process in EFC, indicating net consumption of O$_2$ and production of CO$_2$ (Fig 5).

EFC was metabolically more active from late spring to early fall (from May to October, Fig 6). During this period, GPP showed a stronger diel signal with a higher peak, and ER was also higher than in the winter season. For instance, in summer, the highest 5-minute GPP was 0.98 mg O$_2$ L$^{-1}$ d$^{-1}$, whereas 0.41 mg O$_2$ L$^{-1}$ d$^{-1}$ was the highest rate in winter. The median 5-minute ER was -0.14 mg O$_2$ L$^{-1}$ d$^{-1}$ in summer but -0.031 mg O$_2$ L$^{-1}$ d$^{-1}$ in winter. Similarly, DO also

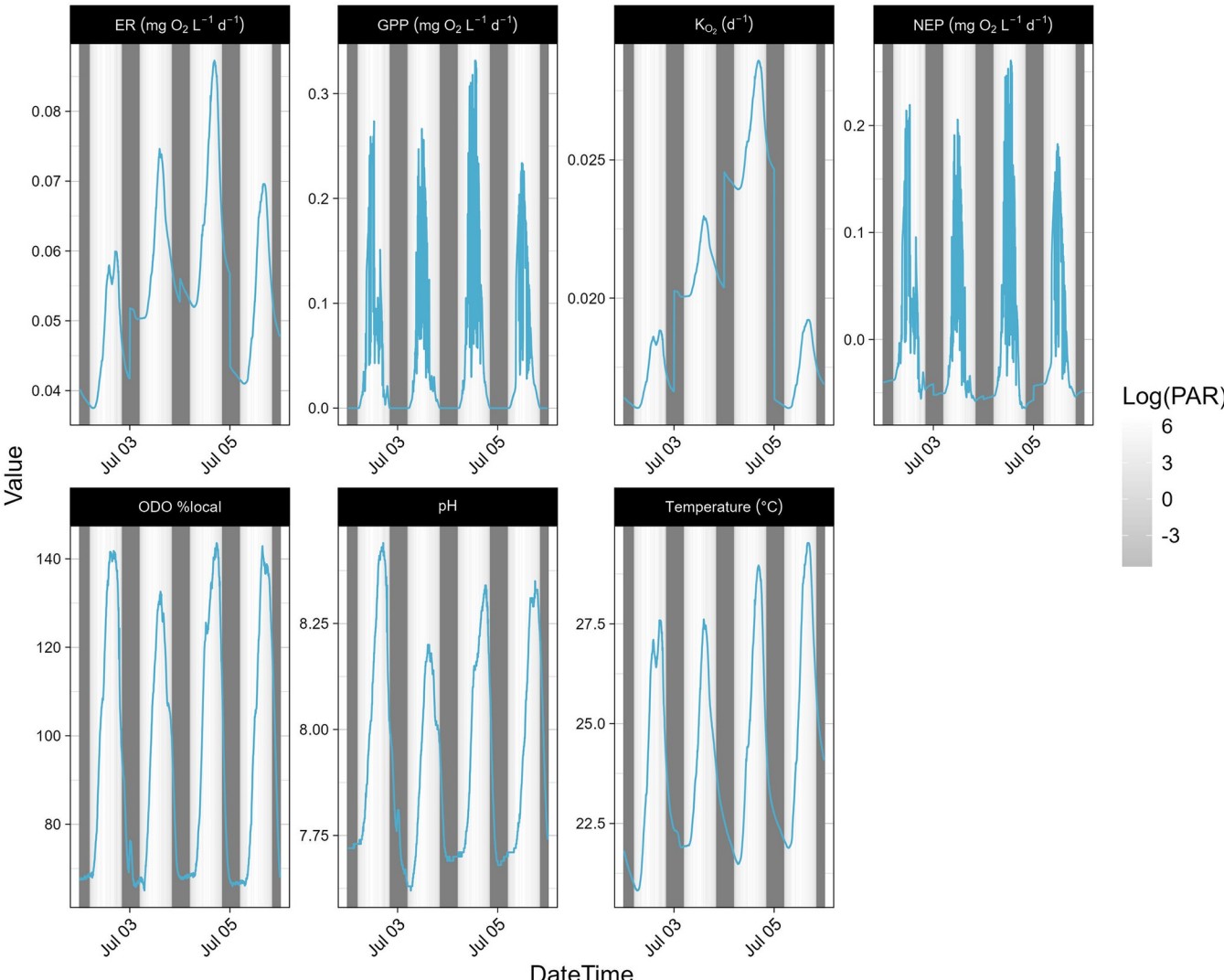

**Fig 3. Time Series for continuous measurements from EXO$_2$ at EFC site 1.** The measurements were made every five minutes from July 2$^{nd}$ to 6$^{th}$, 2022. The shading represents the measured light intensity in log(umol m$^{-2}$ s$^{-1}$). "ODO_per_local" is optical dissolved oxygen percent local saturation.

showed a weaker diel signal in winter. However, DO showed an opposite seasonality, with median values of 7.3 mg L$^{-1}$ for summer and 10 mg L$^{-1}$ for winter because O$_2$ solubility increases with decreasing temperature (Fig 6).

## Effects of temperature and light intensity

The model from Metabolic Theory of Ecology provides the linear expression for the natural log of ER:

$$\ln(\mathrm{ER}) = -\frac{\mathrm{E_A}}{\mathrm{k}}\frac{1}{\mathrm{T}} + \ln(\mathrm{b_0 M^{\frac{3}{4}}}) \tag{2}$$

where M is organism mass, b$_0$ is a constant independent from body size or temperature, E$_A$ is the activation energy, T is the absolute temperature, and k is the Boltzmann constant [28]. In EFC the measured temperature ranged from 1°C to 30°C, and 5-minute ER ranged from

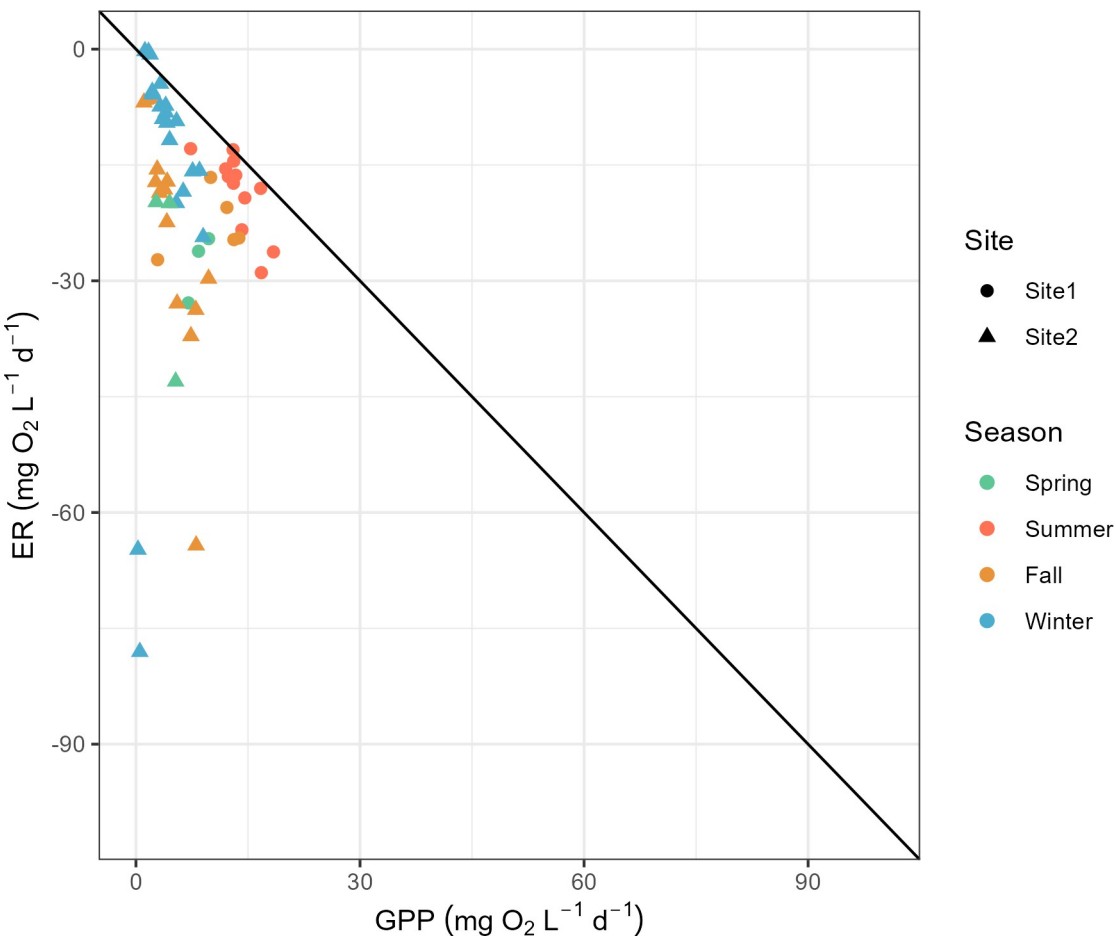

**Fig 4. BASEmetab model estimates of daily values of GPP and ER in East Fork Creek from October 2022 to December 2023.** The dark line is the 1:1 line for GPP = ER.

0.00076 mg $O_2$ $L^{-1}$ $d^{-1}$ to 0.33 mg $O_2$ $L^{-1}$ $d^{-1}$. It was found that for every daily set of 5-minute measurements in EFC there was a negative linear relationship between the inverse of temperature and the natural log of ER that was statistically significant ($p < 0.05$) (Fig 7A). Higher ER rates were associated with higher temperatures. The slope remained consistent throughout the year, indicating that the activation energy for ER does not vary and there is no obvious temperature threshold for ER within the measured temperature range in EFC.

After removing all the data points with zero GPP, a similar relationship was found between temperature and maximum and minimum GPP values (Fig 7). The wide range of GPP values at a given temperature was due to daily variations in PAR (Fig 7). In spring, summer, and fall, the maximum 5-minute GPP values are similar, but in winter, the maximum 5-minute GPP values are smaller (Fig 7). Minimum 5-minute GPP values show a decreasing trend from summer to spring to fall to winter. The highest 5-minute GPP measurement occurred in September with a value of 0.53 mg $O_2$ $L^{-1}$ $d^{-1}$ and the lowest 5-minute GPP measurement occurred in winter with a value of 1.1 x $10^{-7}$ mg $O_2$ $L^{-1}$ $d^{-1}$. Five-minute GPP were linearly correlated with PAR, although the slope varies among seasons (Fig 7C). In general, within a day, the highest 5-minute GPP rate occurred when PAR was highest. Our data show that most variation in GPP is explained by temperature and PAR, as made clear by using multiple linear regression

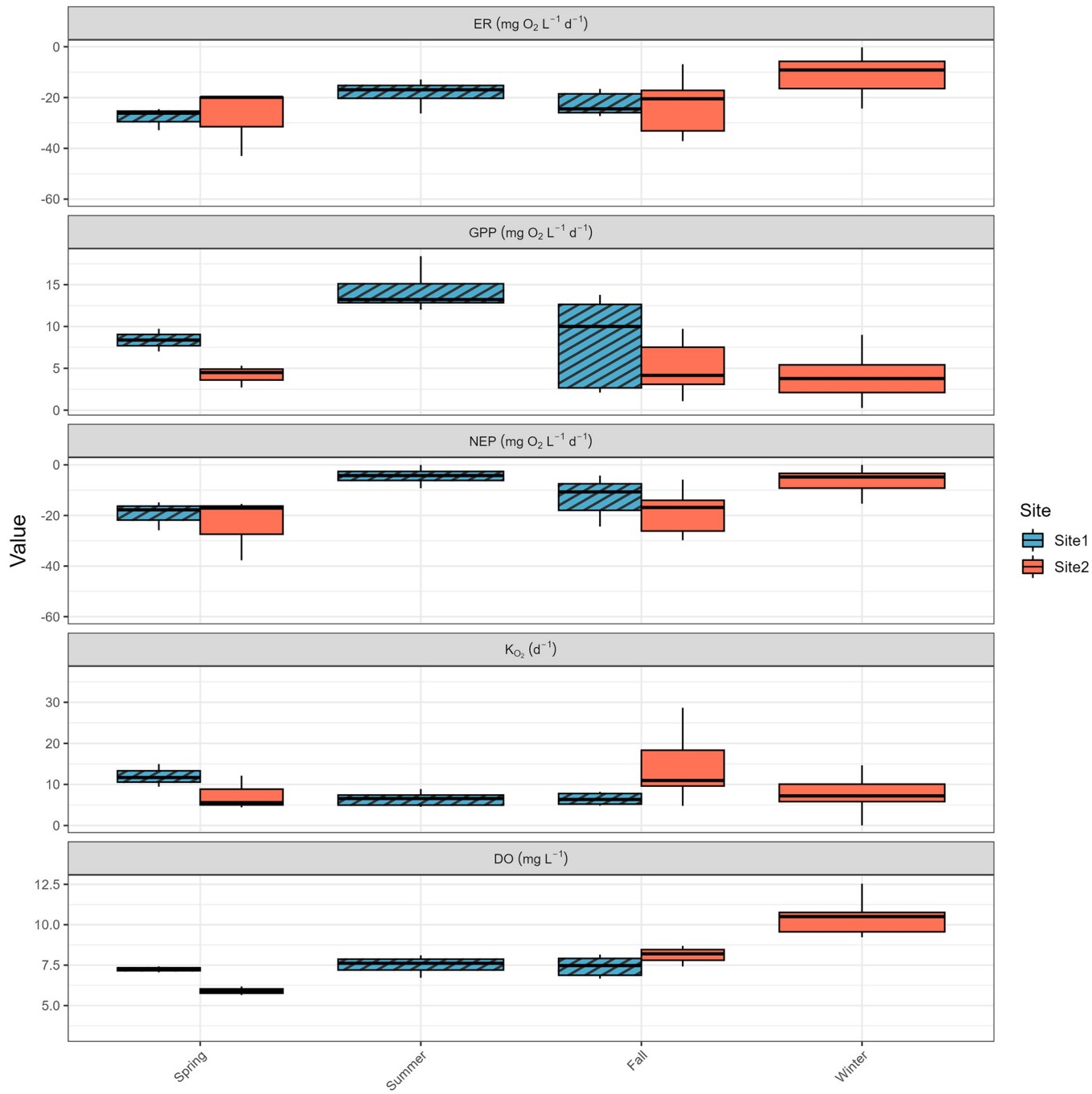

**Fig 5. Seasonal boxplots for BASEmetab average daily values of ER, GPP, NEP, $K_{O2}$, and DO.** Error bars are ±1σ.

on all instantaneous values to obtain Eq (3) (multiple r = 0.74):

$$GPP = 0.000734*T + 0.000321*I \tag{3}$$

where GPP is in mg $O_2$ $L^{-1}$ $d^{-1}$, temperature T is in Celsius, and photosynthetically active radiation I is in μmol $m^{-2}$ $s^{-1}$.

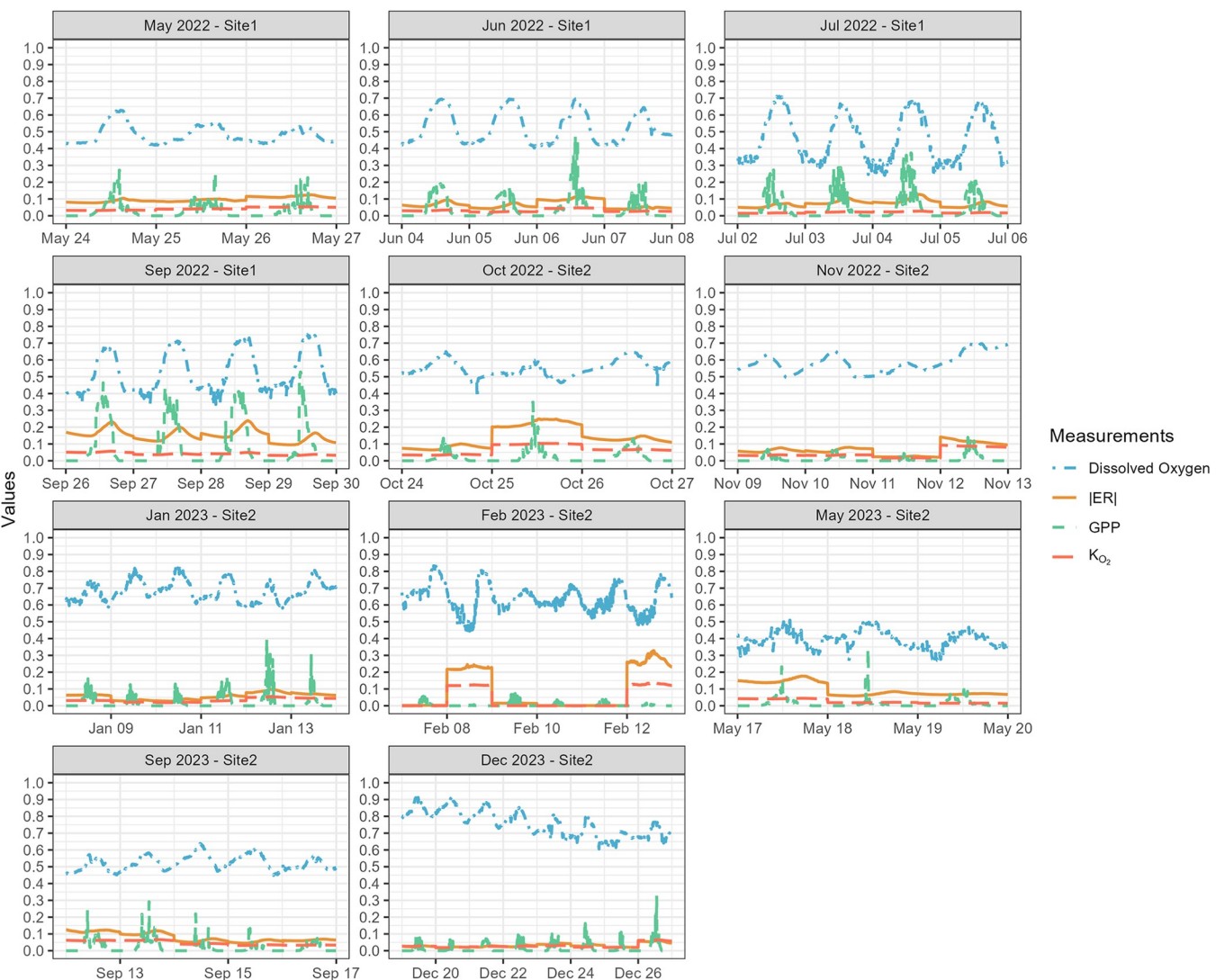

**Fig 6. Time series for 5-minute average EXO2 DO measurements (right y-axis scale) and BASEmetab estimated 5-minute values of |ER| and GPP (mg O$_2$ L$^{-1}$ d$^{-1}$) and K$_{O_2}$ (units of d$^{-1}$) at East Fork Creek.** Summing the 5-minute values for a day gives the total daily values for |ER|, GPP, and K$_{O_2}$. Month and site number are provided in the title of each subplot.

## Dissolved oxygen and hypoxia

The scaling model of stream DO concentrations established by Abdul-Aziz & Gebreslase (19) [29] shows that:

$$DO = 10^{18.94} \cdot T_w^{-7.46} \cdot pH^{0.45} \tag{4}$$

where DO is dissolved oxygen concentration (mg/l) and T$_w$ is water temperature (K). Comparing the model-predicted values and the measured DO concentrations from July 2nd to July 5th, 2022, we found a strong correlation (R$^2$ = 0.76) between measured and predicted concentration but not a 1:1 relationship (S6.1 Fig in S6 File). Our measured temperatures ranged from 20.8°C to 29.5°C. Higher temperatures were associated with higher measured DO concentrations (Fig 8), which is opposite from the trend observed in the seasonal variation of DO

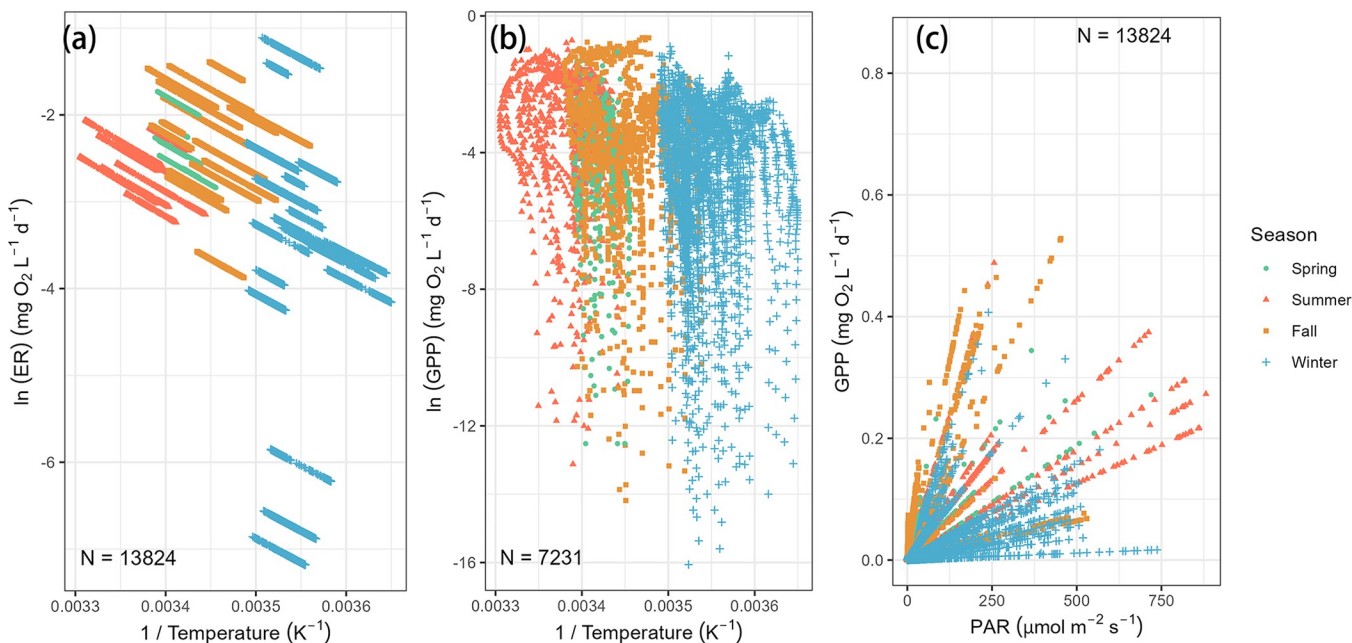

**Fig 7. The relationship between East Fork Creek stream metabolism and its thermal and light regime by season.** (a) The dependence of the natural log of 5-minute |ER| values on 1/T. (b) The dependence of the natural log of 5-minute GPP values on 1/T. (c) The dependence of instantaneous GPP on PAR.

concentration (Fig 5). It was also found that the diurnal variations in measured DO concentrations and equilibrium DO (measured DO / ODO % local) at saturation are anti-phase (Fig 9). The measured DO concentrations reached the daily maximum during daytime when PAR measurements were high and the daily minimum values occurred during the night. The equilibrium DO concentrations showed the opposite diurnal cycle but with a smaller range of change. During the daytime, water can be supersaturated when the measured DO concentration exceeded the saturation concentration.

Hypoxia occurs when the DO concentration is lower than 2 mg L$^{-1}$ [6]. For all DO concentration measurements in EFC, the maximum value of 12.6 mg L$^{-1}$ occurred at 5:50 PM Feb 7th, 2023 (winter) and the minimum value of 3.6 mg L$^{-1}$ occurred at 4:14 AM July 4th, 2023 (summer). The daily DO amplitude (Maximum DO–Minimum DO) has a larger value in summer compared with the other three seasons (Fig 10). The average value of $K_{O2}$ is 9.8 d$^{-1}$. The stream will become vulnerable to hypoxia events when the sites have low $K_{gas\_transfer}$ (< 10 m d$^{-1}$, obtained by multiplying $K_{O2}$ by stream depth in meters) and very negative NEP (< -10 mg O$_2$ L$^{-1}$ d$^{-1}$), and the risk increases for small streams at high temperatures [6]. In EFC, there are 6 days in spring, 1 in summer, 15 in fall, and 5 in winter falling in the region of risk (Fig 11), but based on the field measurements of DO no hypoxia event was observed.

## Discussion

The long-term monitoring of stream metabolism at EFC enables the observation of the cumulative results of environmental changes on both GPP and ER. Using the current results, we explore three river ecosystem drivers susceptible to global change including temperature, light, and organic carbon subsidies [30]. The impact of changes in flow regime and nutrients on stream metabolism is not addressed.

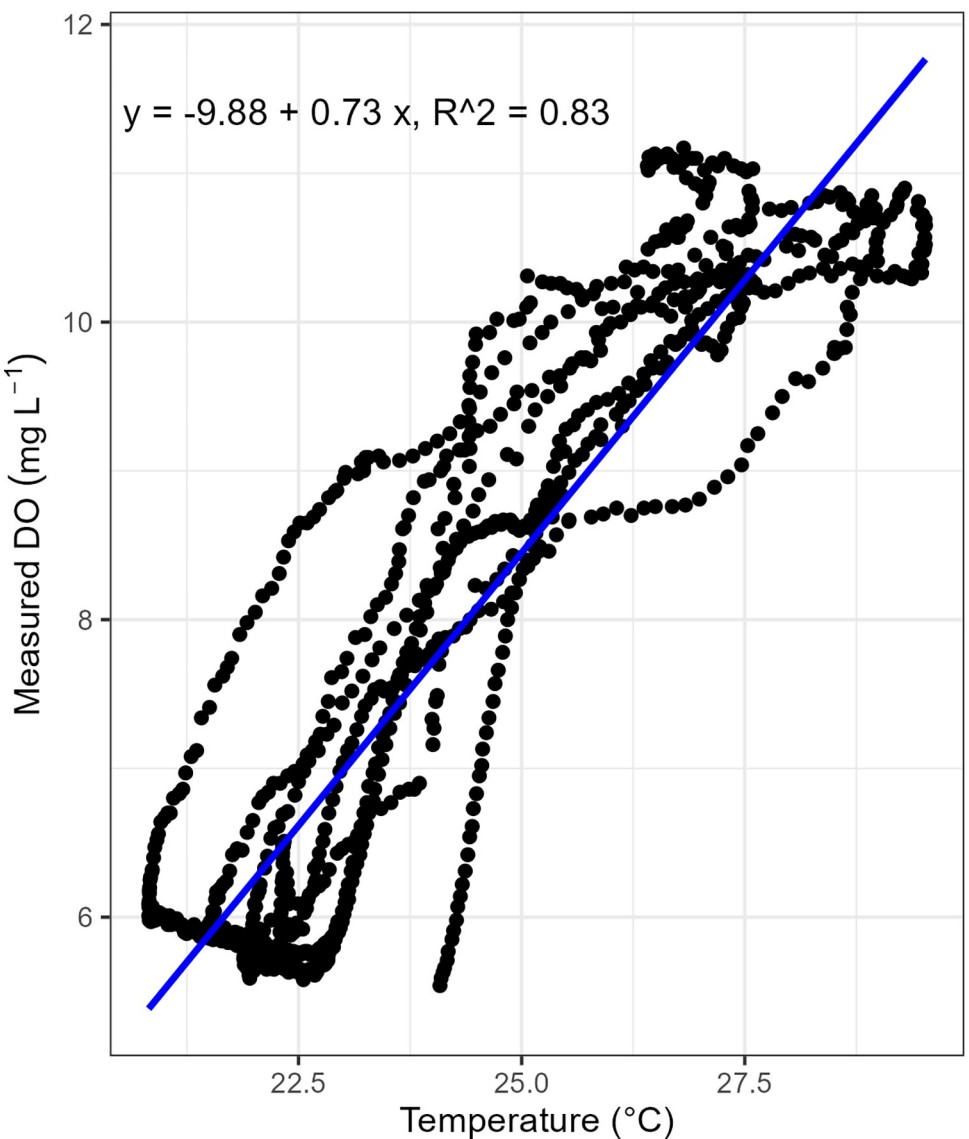

**Fig 8. Temperature vs measured DO concentration.** Measurements were made from July 2nd to July 5th, 2022 at EFC site 1. The blue line is the best linear fit for the datapoints with the equation at the top of the graph.

### Diel chemical cycles

The diel signals in Ca, Mg, and pH (Fig 2) are closely related to the temporal patterns of dissolved oxygen which is dependent on solar radiation, temperature, and biological activity (Fig 3). Due to photosynthesis, the concentration of dissolved oxygen increases during the day. At nighttime respiration becomes the dominant biological process, producing excess $CO_2$ that dissolves in water to form carbonic acid, which releases H+ and decreases the pH. The lower pH during the night (Figs 2 and 3) allows the carbonate minerals calcite and dolomite to redissolve in the water [31], leading to increased concentrations of Ca and Mg (Fig 2).

### Gross primary productivity

As expected, the diurnal variation of GPP is dominated by the availability of PAR (Fig 7), since it is driven by the photosynthetic activity of the benthic algae population. Continuous

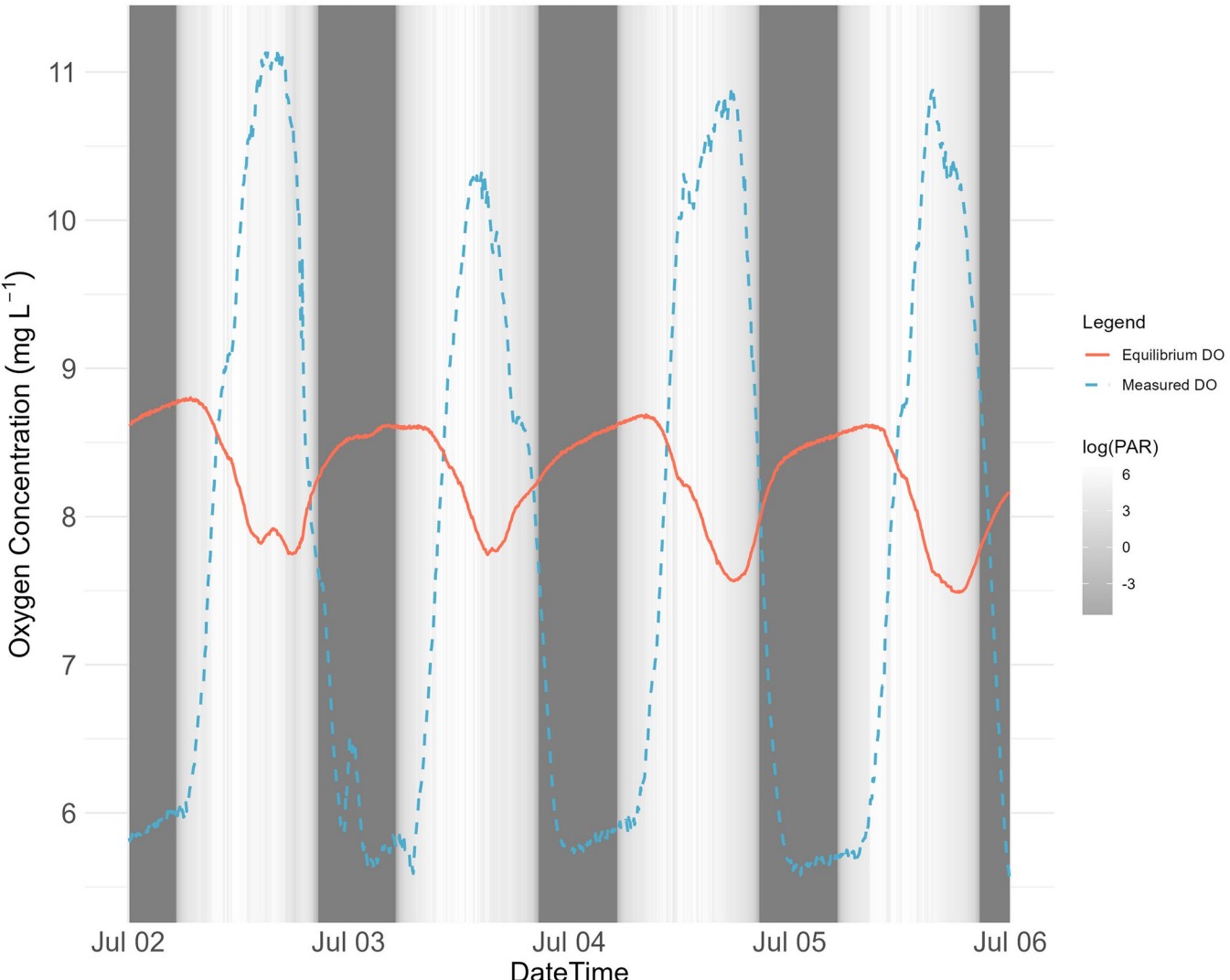

**Fig 9. Time series plot for measured DO concentrations and equilibrium (saturation) DO concentrations from July 2nd to July 5th, 2022 at EFC site 1.** The shading represents the measured illuminance in lux converted to PAR in log($\mu$mol m$^{-2}$ s$^{-1}$).

measurements of GPP and PAR show a stronger correlation than any other parameters (Spearman's r = 0.9, p = 0, S5.2 Fig in S5 File). Similar relationships between GPP and PAR were found in many other streams. Measurements taken from 11 streams in north Florida found a linear relationship between mean benthic light and mean GPP with R$^2$ = 0.8 [32]. Measurements in an urban stream containing wastewater treatment plants found that light intensity was the dominant control on GPP [33]. In three forested British Columbia streams, artificial light addition was found to increase the GPP by at least 31%, and artificial shading was found to decrease the GPP by 11% [34].

However, for seasonal variations of GPP, PAR is no longer a dominant factor. A weak relationship between the seasonal variation of GPP and PAR is found in our measurements (Fig 7C). The variations in slopes show that the photosynthetic efficiencies, the amount of extra GPP production per unit increase of PAR, during winter were the lowest among the four seasons. Unexpectedly, photosynthetic efficiencies during spring and fall were higher than

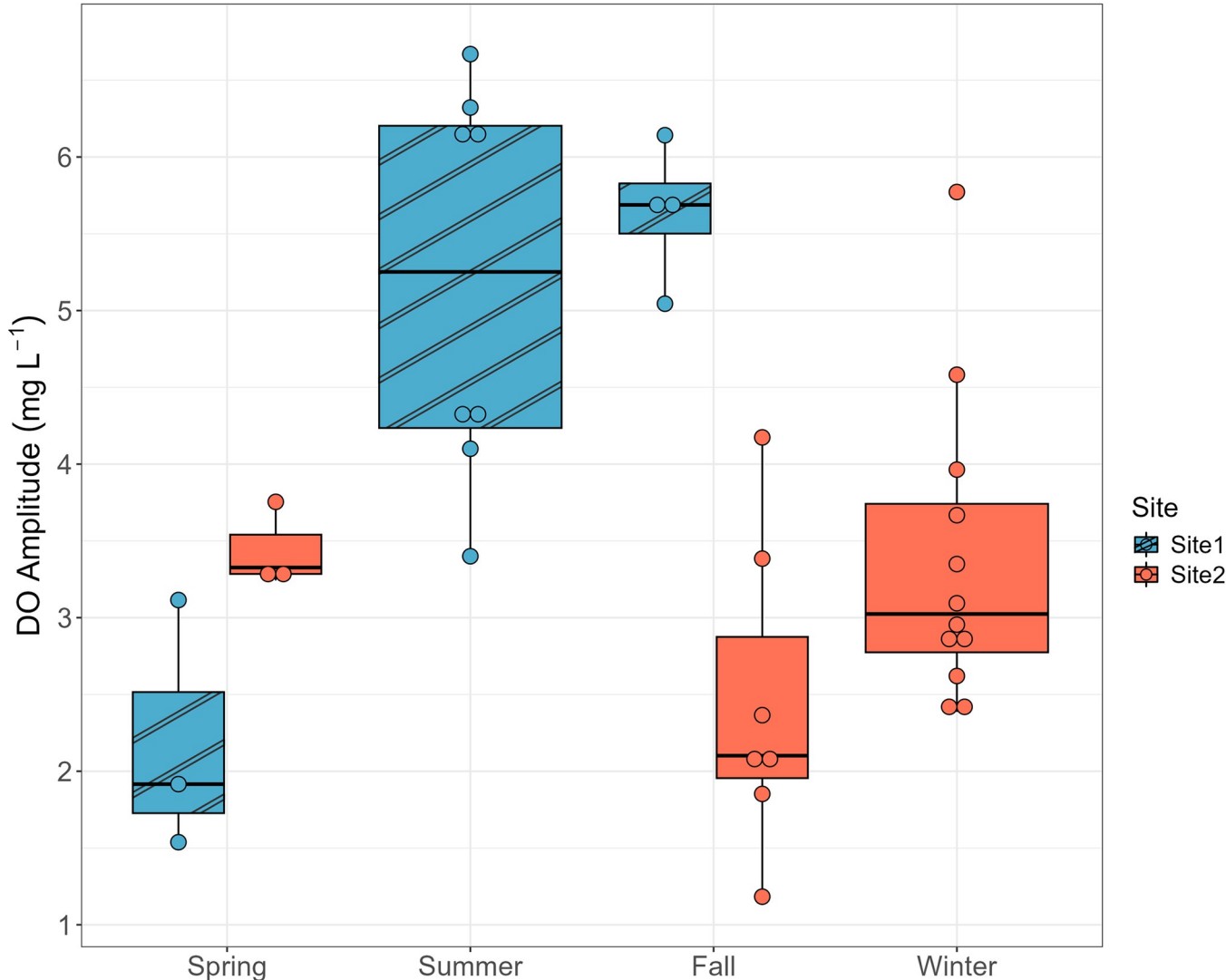

**Fig 10. Boxplot showing the seasonal variations of daily DO amplitude (daily maximum–daily minimum values).**

summer (Fig 7). PAR can be strongly influenced by uncontrollable factors such as weather conditions and leaf coverage, so there are winter days with similar or even higher PAR but with lower GPP than the summer. Several studies have indicated that GPP per unit chlorophyll will no longer increase once PAR is above a certain value [35–37]. No unanimous conclusion has been reached regarding the PAR saturation point since it can vary across seasons and different algae communities, but this maximum productivity limit can be used to explain the weak seasonal relationship between GPP and PAR (Fig 7). The measurements in natural biofilms in two Mediterranean streams show that in shaded streams, seasonality did not alter the maximum GPP per chlorophyll [35]. Low GPP in winter days in EFC could be caused by low benthic algae concentration in the streambed.

Another explanation of the seasonal variations of GPP is the impact of temperature. Temperature increase will lead to a faster photosynthesis rate due to its kinetic effect and increase in the activity of photosynthetic enzymes [38]. Our continuous measurements show a significant but weak correlation between GPP and temperature ($r = 0.51$, $p = 0$). Average daily GPP

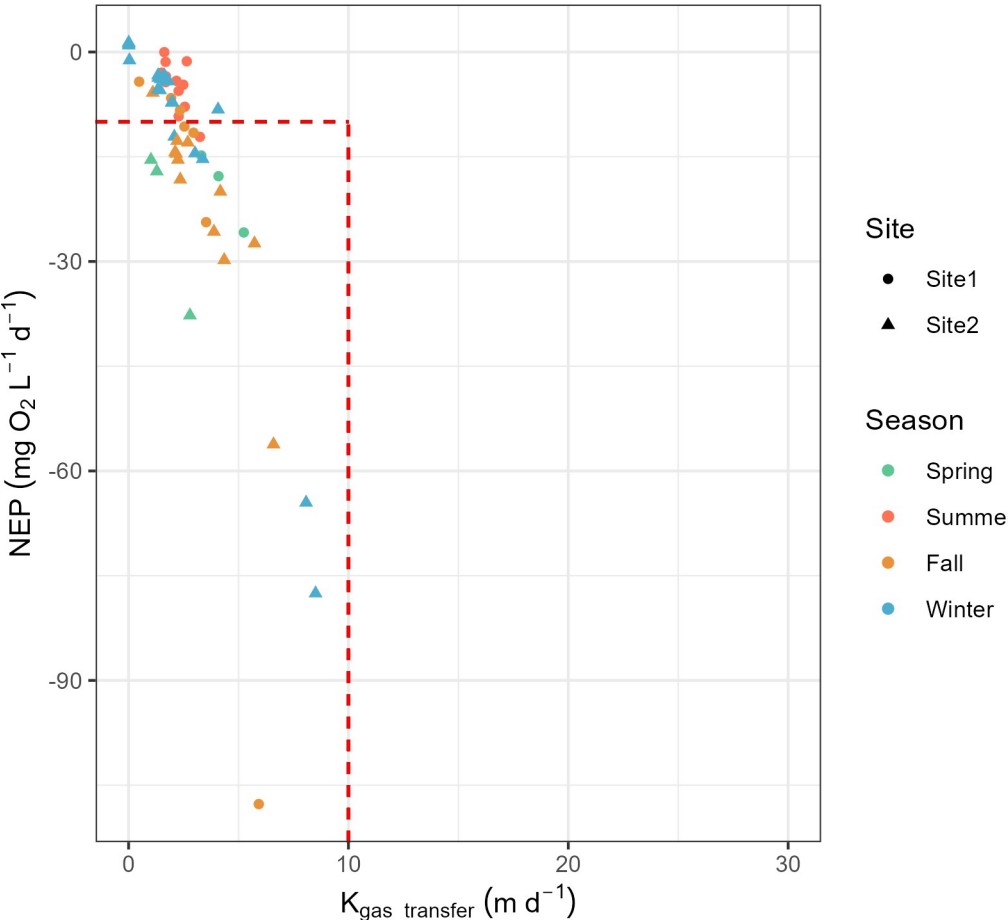

**Fig 11. Measured daily values of the gas transfer coefficient (equal to the reaeration coefficient $K_{O2}$ times stream depth in meters) versus net ecosystem productivity NEP.** Two red segments at NEP = -10 mg $O^2$ $L^{-1}$ $d^{-1}$ and $K_{gas\_transfer}$ = 10 m $d^{-1}$ define a region at the bottom left where streams are vulnerable to hypoxia [6].

shows a seasonal variation pattern that is synchronous to the temperature changes across the four seasons in EFC (Fig 5), and instantaneous GPP had higher daily minimum values on warmer days, but for maximum instantaneous GPP, the temperature effect is not significant (Fig 7). Even though GPP shows a positive correlation with temperature seasonally, diurnal variations of GPP are dominated by light availability. Similar GPP seasonal patterns were found at Akabira River in Japan where GPP ranged from 1.8 to 36.4 g $O_2$ $m^{-2}$ $d^{-1}$ [37]; and at geothermally warmed streams in south-west Iceland where GPP ranged from 2 to 28 g $O_2$ $m^{-2}$ $d^{-1}$ [39]. At Diamond Creek in Colorado, GPP gradually increased from winter to summer, but a flooding event caused a decline in August due to scouring of benthic algae, indicating the impact of stream dynamics on GPP [40].

## Ecosystem respiration

The magnitude of ER exceeded GPP on 93% of measured days. EFC is usually heterotrophic at our field sites because there is abundant leaf litter. The instantaneous measurements in EFC show that higher temperature will lead to a faster respiration rate (r = 0.8, S5.2 Fig in S5 File), which aligns well with the common recognition of the temperature dependence of ER. Compared with the terrestrial system, freshwater respiration is more dependent on temperature

with an activation energy of around 0.65 eV, whereas terrestrial activation energy is around 0.32 eV [10]. In EFC, no temperature threshold was observed for ER and the activation energy (proportional to the slope in Fig 7) remains consistent throughout the four seasons. Different patterns were observed in terrestrial systems where ER activation energy drops when the soil temperature is above 11.4˚C [41]. It is also usually found in both model estimates and empirical measurements that ER is more sensitive to temperature variations than GPP [37,42,43]. Hence, the temperature rise caused by climate change can make headwater streams more heterotrophic and emit more $CO_2$ into the atmosphere.

However, temperature dependency does not fully account for the seasonal variation of ER in EFC. Despite the large difference between mean temperatures in summer and winter in middle TN, winter |ER| is slightly lower than summer |ER|. Peak |ER| occurred in the fall at lower mean temperature than the summer. This deviation from predictions of the metabolic theory of ecology can be explained by changes in availability of labile organic carbon. A similar increase of |ER| in fall was found in TN and NC forested streams, and the study concluded that an increase in leaf litter inputs leads to this fall |ER| peak [44]. Other natural variations such as changes in the river network geomorphology [45] and depletion of autochthonous carbon [46] were also found as limiting factors of ER.

Our results suggest that ER variations in EFC can be explained by interactions between the stream and its surrounding environment. EFC, a headwater stream near a perennial forest, should not be treated as a closed system whose thermal regime dominates the ER. Instead, seasonal variations in allochthonous carbon may have a larger influence on ER than temperature. The decomposition of leaf litter in EFC increases the demand for dissolved oxygen and leads to a peak in |ER| in the fall (Fig 5). The slightly lower |ER| values in winter and summer are likely caused by the combined results of lower temperatures and organic carbon limitation.

## Dissolved oxygen

The DO concentration in EFC is controlled by both solubility and stream metabolism. Variations in solar radiation drive the seasonal temperature changes that affect oxygen solubility. As a result, average DO concentrations are higher in winter (Fig 5). The impact of metabolic activities becomes more obvious when looking at the pattern of diurnal DO variations. During the summer season when the metabolic activity is higher than in other seasons, the magnitude of diurnal DO change is larger due to the higher peak primary productivity (Figs 6 and 10). During the day when temperature increases, the equilibrium DO concentration at saturation decreases, but the measured DO concentration increases due to active oxygen production, causing supersaturation (Fig 9). In EFC, the dominant effect of metabolic activities on diurnal DO variation also causes the deviation from the temperature- and pH-dependent scaling law proposed by Abdul-Aziz & Gebreslase [19,29]. Within a 24-hour day, DO shows a positive relationship with the temperature since the decrease in solubility during the daytime is overwhelmed by the active oxygen production (Fig 9).

Due to turbulence and active oxygen mixing in headwater streams, EFC shows an average reaeration coefficient $K_{O2}$ of 9.8 d-1 (Fig 5) [47]. Higher values of $K_{O2}$ and $K_{gas\_transfer}$ should decrease the risk of hypoxia by increasing the rate of replacement of DO consumed by respiration (Fig 11). However, our results still show a risk of hypoxia events in EFC when ER is high and oxygen cannot be supplied by either oxygen dissolution (reaeration) or primary production quickly enough to replace the oxygen consumed by respiration (Fig 11).

**Model limitations.**   Since we carefully checked the EFC's compatibility for a single-station estimation and performed the quality checks for the model output, our estimates of the stream metabolism parameters GPP, ER, and $K_{O2}$ should accurately represent the stream's metabolic

regime throughout the entire year. However, the parameterization choices for GPP, represented by the term $AI_i^p$, and ER, represented by the term $R(\theta^{(T_i - T)})$, in the model (Eq 1) may affect our observation of the relationships between EFC metabolic regimes and environmental conditions.

This parameterization choice offers a computationally efficient way to estimate stream metabolism using Bayesian model fitting. On the other hand, it simplifies the factors that determine GPP and ER in the real world. Therefore, in the model outputs, we found perfect linear ER-temperature (Fig 7A) and GPP-PAR (Fig 7C) relationships, but the true relationship can be more complicated. Based on the model, in some summer days, even though PAR values are high, GPP does not increase proportionally, leading to a lower value of A, the primary production per quantum of light. On the GPP-PAR diagram (Fig 7C), this lower A value shows up as a shallower slope compared to days in fall and spring. However, the estimated daily values for GPP and ER are still valid and allow us to make reliable observations for the seasonal variations in stream metabolism.

## Conclusions

As a rural headwater stream, EFC provides a perfect opportunity to observe how stream metabolism responds to diurnal and seasonal environmental variations without human disturbances. Our results suggest that EFC remains heterotrophic regardless of seasonal variations. The most negative NEP occurs in the fall when leaf litter enters the stream and boosts ER through organic decomposition. GPP shows a strong seasonal variation with a high peak in summer, which can be a combined result of higher temperatures, higher PAR levels, and possibly an increase in benthic producer population. On the other hand, daily GPP and PAR show a synchronous variation, indicating the dominant effect of light over GPP. The seasonal variation in DO is mainly driven by the temperature dependence of oxygen solubility, whereas diel variations in DO are dominated by the stream's metabolic activity, which aligns well with the observation for diurnal GPP variations. This diel DO signal does not follow the temperature- and pH-dependent scaling law proposed by Abdul-Aziz & Gebreslase [29] (19) because the latter does not incorporate biological effects (except indirectly through pH changes). No hypoxia event was observed, but we observed conditions of increased hypoxia risk in all four seasons when ER was high and oxygen could not be supplied quickly by either oxygen dissolution (reaeration) or primary production. Our work shows that headwater streams can be metabolically more active when temperature is higher, with higher GPP and ER rates and a larger daily range of DO concentrations. However, ER can be more susceptible to changes in carbon inputs than temperature changes.

While this study provides insights into factors affecting headwater stream metabolism, the long-term impact of climate change on oxygen and carbon dynamics in headwater streams is still uncertain. Further research is needed to understand how long-term environmental changes will affect stream metabolism. Future studies of metabolism in headwater streams could reduce uncertainties in model parameters by increasing the number of sample sites to assess along-stream variability, characterizing the composition and role of the benthic algae population, measuring $CO_2$ emissions, and measuring seasonal variations in carbon inputs.

## Supporting information

**S1 File. Sensor precision tests.** S1.1 MiniDOT vs EXO2. S1.2 HOBO Light Logger vs LI-1500 Light Sensor Logger. S1.3 HOBO Light Logger Orientation. S1.4 Consistency of MiniDOT Measurements.
(DOCX)

**S2 File. Site result comparisons.**
(DOCX)

**S3 File. Water chemistry, July 2022.**
(DOCX)

**S4 File. Time series plots for July 2022 discrete measurements.**
(DOCX)

**S5 File. Correlation matrices.**
(DOCX)

**S6 File. Measured DO vs predicted DO.**
(DOCX)

**S7 File. Sample BASEmetab output for July 4, 2022 at East Fork Creek site 1.**
(DOCX)

**S8 File. Instructions for stream metabolism measurements.**
(DOCX)

## Acknowledgments

We thank Weizhuo Jing, Jialei Wei, Gabriel Perez, and Ella Daugherty for help in the field and laboratory, and Jonathan Oppenheimer and Lee Anne O'Brian from the Center for Sustainable Stewardship for their support.

## Author Contributions

**Conceptualization:** John C. Ayers.

**Data curation:** Ming Chen, John C. Ayers.

**Formal analysis:** Ming Chen, John C. Ayers.

**Funding acquisition:** Ming Chen.

**Investigation:** John C. Ayers.

**Methodology:** Ming Chen, John C. Ayers.

**Project administration:** John C. Ayers.

**Resources:** John C. Ayers.

**Software:** Ming Chen, John C. Ayers.

**Supervision:** John C. Ayers.

**Validation:** Ming Chen, John C. Ayers.

**Visualization:** Ming Chen, John C. Ayers.

**Writing – original draft:** Ming Chen.

**Writing – review & editing:** John C. Ayers.

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
