## [Decision Letter · Decision Letter 0]

27 Sep 2024

PONE-D-24-33784Environmental drivers of stream metabolism in a Middle TN headwater streamPLOS ONE

Dear Dr. Ayers,

Thank you for submitting your manuscript to PLOS ONE. After careful consideration, we feel that it has merit but does not fully meet PLOS ONE’s publication criteria as it currently stands. Therefore, we invite you to submit a revised version of the manuscript that addresses the points raised during the review process.

We look forward to receiving your revised manuscript.

Kind regards,

Nicholas D Ward

Academic Editor

PLOS ONE

Journal Requirements:

1. When submitting your revision, we need you to address these additional requirements. Please ensure that your manuscript meets PLOS ONE's style requirements, including those for file naming. The PLOS ONE style templates can be found at https://journals.plos.org/plosone/s/file?id=wjVg/PLOSOne_formatting_sample_main_body.pdf and https://journals.plos.org/plosone/s/file?id=ba62/PLOSOne_formatting_sample_title_authors_affiliations.pdf 2. Please note that PLOS ONE has specific guidelines on code sharing for submissions in which author-generated code underpins the findings in the manuscript. In these cases, we expect all author-generated code to be made available without restrictions upon publication of the work. Please review our guidelines at https://journals.plos.org/plosone/s/materials-and-software-sharing#loc-sharing-code and ensure that your code is shared in a way that follows best practice and facilitates reproducibility and reuse. 3. Please include your full ethics statement in the ‘Methods’ section of your manuscript file. In your statement, please include the full name of the IRB or ethics committee who approved or waived your study, as well as whether or not you obtained informed written or verbal consent. If consent was waived for your study, please include this information in your statement as well. 4. Please include captions for your Supporting Information files at the end of your manuscript, and update any in-text citations to match accordingly. Please see our Supporting Information guidelines for more information: http://journals.plos.org/plosone/s/supporting-information.

Reviewers' comments:

Reviewer's Responses to Questions

**Comments to the Author**

1. Is the manuscript technically sound, and do the data support the conclusions?

Reviewer #1: Partly

Reviewer #2: No

2. Has the statistical analysis been performed appropriately and rigorously? 

Reviewer #1: Yes

Reviewer #2: No

3. Have the authors made all data underlying the findings in their manuscript fully available?

Reviewer #1: Yes

Reviewer #2: Yes

4. Is the manuscript presented in an intelligible fashion and written in standard English?

Reviewer #1: Yes

Reviewer #2: No

5. Review Comments to the Author

Reviewer #1: General comments:

This manuscript describes an experimental study investigating headwater stream metabolic

regimes in East Fork Creek, a headwater stream in middle Tennessee. The manuscript is generally well written, and the methods are well described, but I do not believe in this state, the results are novel or transformative enough for PLOS One. I provide some other comments that I hope help the authors streamline the manuscript.

Introduction:

-The first three paragraphs of the introduction describe very high-level general knowledge on stream metabolism. This is unnecessary and can be summarized in a couple of sentences with the appropriate citations. Instead, the introduction should synthesize what we know about how stream metabolism is affected by the controls the authors are interested in testing (e.g., temperature; light intensity, etc) and what knowledge gaps this work is contributing to.

Materials and Methods:

-The method described to measure flow velocity is inappropriate for this kind of study, I suggest eliminating the comment.

-The sensor precision test contributes little to the manuscript. I suggest removing the section and adding a line under field campaign setup along the lines of : “We confirmed the MiniDOTs and 153 HOBO Pendant MX Temperature/Light Data Loggers serve well for the purpose of this study (XXX in Supporting Information)”.

-If Site 1 was abandoned it should be simply removed from the study and the Site results comparison section eliminated.

Results

-For the section describing seasonal variations, there is many data gaps in figure 4 and the data would be better presented in boxplots separated by season.

General

-Address inconsistencies in significant figures in values throughout the manuscript and in figures. Also, in the text, values of ER, GPP, etc are written with too many significant figures.

Reviewer #2: General comments: This paper aims to investigate seasonal and diurnal variations in metabolism parameters on a headwater stream. To do this the authors measure key environmental parameters at 3 locations within the East Fork Creek. They estimate metabolism utilizing dissolved oxygen data from sensors. The manuscript in its current state requires major revisions.

The current writing of the manuscript includes several results and pieces of information that are not contributing to the main story of the paper and thus the ideas behind these extra pieces of information are not fully developed throughout. For example, lines 147-149 mention “Another goal of this study was to test the precision and reliability of new light and dissolved oxygen sensors and see if they can make continuous stream metabolism measurements at a lower cost”, this goal is not directly contributing to the original goals stated on the introduction, if a relationship exists it is not clear. Furthermore, results from the metabolic ecological model as well as scaling from equation 4 are not fully folded into the discussions which makes the reader question the need for this extra analysis. Additionally, the authors describe 3 key sites for their study but then it seems that the major results are only from one site, it is not clear the need for all the sites or which results come from which of the riverine locations.

Overall, the manuscript readability needs to be significantly improved. I recommend thinking deeply about the goals and questions to explore on this manuscript and ensure that all of the results and data presented are clearly addressing those goals and questions. It would perhaps help to have a combined results and discussion sections with sub-headers that delineate the connection of each section to the goals and questions.

Finally, I have major concerns about how metabolism in the lake was calculated as well as the discrepancy in orders of magnitudes for metabolism metrics across figures. With the current state of the writing, I was not able to discern if there is a justifiable reason for the difference in these magnitudes or if a lake specific methodology was applied for lake metabolism calculations.

Specific comments:

42-47 I recommend reviewing Chapter 34 - Stream Metabolism (https://doi.org/10.1016/B978-0-12-813047-6.00012-7) and adjusting equations accordingly.

54-63 Discuss GPP controls in detail, add similar content for ER and NEP.

Please revise the statements in 69-79 and back them up with literature. This paragraph is very low on citations. I am familiar with metabolism literature, and I believe some of the statements made are not supported by published studies. It seems the authors are conflicting metabolism studies (i.e., studies that estimate ER and GPP via measured diel variations of dissolved oxygen in water) and freshwater studies that measure CO2 concentrations.

95 Seems that sites downstream of the lake might be 2nd order streams.

162 Were samples filtered upon returning to the university? If so, how many days did the samples spend not filtered?

188-192 Results, rephrase please.

223-239 It is unclear why the authors need to estimate metabolism at 2 sites. Seems the questions posed on the instruction could be addresses just by looking at the one station metabolism outputs from Site 2 since it had more robust data and data collected for a longer period of time.

230 How was metabolism at Stephens Lake estimated? A lake doesn’t meet the assumptions for BASEmetab.

235-236 “instantaneous values of GPP, ER, and K at the two sites are highly correlated (Pearson and Spearman p values always < 0.05 for the null hypothesis that the correlation is zero)” This statement needs reporting of correlation coefficients in addition to p-value. The correlation might be significant but not necessarily strong.

244-247 Unclear on the connection of these sentences with the 242-244 statements.

250 Is this at site 2? Also, were the DO, HOBO and PAR sensors deployed for 57 days or just 3-7 days as stated in line 128?

Please include plots of the time series of the DO data in the SI.

Was discharge measured? Since the sites are downstream of the lake, what is the influence that the lake has on the stream discharge and therefore the apparent seasonal patterns?

Fig 4. Methods need to include descriptions on average calculations or state if this is a model output.

ER values across multiple figures are positive. ER represents the consumption of DO for which is usually negative, is this the absolute value of ER? If so, please add this information to the methods and the Figure caption.

285-291 Unclear what was done here. What do the authors mean by “caused by the BASEmetab design” and what do the mean with “calculation does not include any information from the previous day”. A feature of the Bayesian modeling is that you rely on information from the previous days to estimate subsequent days.

293 Is this site 1 or site 2?

Fig 5: Figures should be placed within the section they are referenced. Was the DO time series data QAQC? If so what type of rules were followed? Metabolism estimates plotted on Fig 5 seem orders of magnitude lower than the averages on Fig 3 and 4, how are they different?

Fig 6, what are the units for Ca, Mg and DIC measurements?

307-309 Needs correlation coefficients and p-values in text.

460 Reference style doesn’t match.

The discussion doesn’t address all the results presented in the results section. For example, results from the metabolic theory of ecology or the scaling results from equation 4 are not addressed. It is also not clear how the discussion is answering the questions stated on the introduction as the goal of the paper and how those answers help to “better understand the carbon budget for headwater streams in a future with climate change”.

6. PLOS authors have the option to publish the peer review history of their article (what does this mean?). If published, this will include your full peer review and any attached files.

Reviewer #1: No

Reviewer #2: No

---

## [Author Response · Author response to Decision Letter 0]

7 Nov 2024

We thank the reviewers for their helpful suggestions to improve the manuscript. We have made extensive revisions to the manuscript to address the reviewer’s concerns, making the text more focused and making clear which site measurements were made at. All figures were revised, and Figure 1 was replaced with a map based on public domain data from the USGS.

---

## [Editor Report · Decision Letter 1]

12 Nov 2024

PONE-D-24-33784R1Environmental drivers of stream metabolism in a Middle TN headwater streamPLOS ONE

Dear Dr. Ayers,

Thank you for submitting your manuscript to PLOS ONE. After careful consideration, we feel that it has merit but does not fully meet PLOS ONE’s publication criteria as it currently stands. Therefore, we invite you to submit a revised version of the manuscript that addresses the points raised during the review process.

The manuscript was sent back to the two previous reviewers who both unfortunately declined to review the manuscript a second time. Rather than start the review process over with two new reviewers, I have carefully read through your response to the reviewers and the revised manuscript draft. You have done a great job revising the manuscript text in response to each major piece of feedback from the reviewers. I have no further suggestions for the text itself but feel that the figure quality can be improved prior to accepting the manuscript. Here are a few edits I would like you to make in your next revision, mostly related to labels:

Figure 2: The labels appear to be the “raw” text likely used as column labels in your data files. Please clean these up by using sub/superscripts and parentheses for units. For example change GPP_mgO2L-1d-1 to GPP (mg O_2_ L^-1^d^-1^). Likewise, capitalize Measurement. You might consider making the fonts larger as well since I needed to zoom in substantially to read all of the labels.

Figure 3: Same comment as above. Modify all of the labels with super/subscripts and parentheses. In this case listing every date on the x axis is a bit busy, consider only labeling every other day.

Figure 4: Use super/subscripts for the labels

Figure 5: Same comment as above.

Figure 6: Same comment as above for y axis label.

Figure 7: Same comment as above for both axis labels.

Figure 8: Same comment as above for y axis label.

Figure 9: For consistency with other figures change mg/L to mg L^-1^. Consider changing “Values” to “O_2_ concentration”

Figure 10: Same comment as for figure 9

Figure 11: Same, please use super/sub scripts

Please pay similar attention to the figures and tables in the supporting information file to ensure that they are as high quality as the main text figures and use consistent style (e.g., superscripts for units instead of mg/L)

We look forward to receiving your revised manuscript.

Kind regards,

Nicholas D Ward

Academic Editor

PLOS ONE

Journal Requirements:

**Additional Editor Comments:**

The manuscript was sent back to the two previous reviewers who both unfortunately declined to review the manuscript a second time. Rather than start the review process over with two new reviewers, I have carefully read through your response to the reviewers and the revised manuscript draft. You have done a great job revising the manuscript text in response to each major piece of feedback from the reviewers. I have no further suggestions for the text itself but feel that the figure quality can be improved prior to accepting the manuscript. Here are a few edits I would like you to make in your next revision, mostly related to labels:

Figure 2: The labels appear to be the “raw” text likely used as column labels in your data files. Please clean these up by using sub/superscripts and parentheses for units. For example change GPP_mgO2L-1d-1 to GPP (mg O2 L-1d-1). Likewise, capitalize Measurement. You might consider making the fonts larger as well since I needed to zoom in substantially to read all of the labels.

Figure 3: Same comment as above. Modify all of the labels with super/subscripts and parentheses. In this case listing every date on the x axis is a bit busy, consider only labeling every other day.

Figure 4: Use super/subscripts for the labels

Figure 5: Same comment as above.

Figure 6: Same comment as above for y axis label.

Figure 7: Same comment as above for both axis labels.

Figure 8: Same comment as above for y axis label.

Figure 9: For consistency with other figures change mg/L to mg L-1. Consider changing “Values” to “O2 concentration”

Figure 10: Same comment as for figure 9

Figure 11: Same, please use super/sub scripts

Please pay similar attention to the figures and tables in the supporting information file to ensure that they are as high quality as the main text figures and use consistent style (e.g., superscripts for units instead of mg/L)

---

## [Author Response · Author response to Decision Letter 1]

25 Nov 2024

We have addressed all issues raised by Editor Nicholas Ward, which were mostly fixing figure labels. In addition, we

1. Updated the caption for Figure 5 by adding DO and KO2.

2. Fixed the label for S2.1 Figure in the supplement, which was incorrectly labeled as S1.2. 

3. In S3.1 Table changed the column names to scientific notation as mentioned by the reviewer to maintain consistency.

---

## [Editor Report · Decision Letter 2]

4 Dec 2024

Environmental drivers of stream metabolism in a Middle TN headwater stream

PONE-D-24-33784R2

Dear Dr. Ayers,

We’re pleased to inform you that your manuscript has been judged scientifically suitable for publication and will be formally accepted for publication once it meets all outstanding technical requirements.

Kind regards,

Nicholas D Ward

Academic Editor

PLOS ONE
---

## [Editor Report · Acceptance letter]

16 Dec 2024

PONE-D-24-33784R2 

PLOS ONE

Dear Dr. Ayers, 

I'm pleased to inform you that your manuscript has been deemed suitable for publication in PLOS ONE. Congratulations! Your manuscript is now being handed over to our production team.

Kind regards, 

on behalf of

Dr. Nicholas D. Ward 

Academic Editor

PLOS ONE